# Inherent potential of steelmaking to contribute to decarbonisation targets via industrial carbon capture and storage

Sicong Tian [1,2], Jianguo Jiang [2,3], Zuotai Zhang[4] & Vasilije Manovic[5]

Accounting for ~8% of annual global $CO_2$ emissions, the iron and steel industry is expected to undertake the largest contribution to industrial decarbonisation. Despite the launch of several national and regional programmes for low-carbon steelmaking, the techno-economically feasible options are still lacking. Here, based on the carbon capture and storage (CCS) strategy, we propose a new decarbonisation concept which exploits the inherent potential of the iron and steel industry through calcium-looping lime production. We find that this concept allows steel mills to reach the 2050 decarbonisation target by 2030. Moreover, only this concept is revealed to exhibit a $CO_2$ avoidance cost (12.5–15.8 €$_{2010}$/t) lower than the projected $CO_2$ trading price in 2020, whilst the other considered options are not expected to be economically feasible until 2030. We conclude that the proposed concept is the best available option for decarbonisation of this industrial sector in the mid- to long-term.

---

[1] School of Engineering, Macquarie University, Sydney, NSW 2109, Australia. [2] Key Laboratory of Solid Waste Management and Environment Safety (Ministry of Education), Tsinghua University, Beijing 100084, P. R. China. [3] School of Environment, Tsinghua University, Beijing 100084, P. R. China. [4] School of Environmental Science and Engineering, Southern University of Science and Technology, Shenzhen 518055, P. R. China. [5] Combustion and CCS Centre, Cranfield University, Bedford, Bedfordshire MK43 0AL, UK. Correspondence and requests for materials should be addressed to S.T. (email: sicong.tian@outlook.com)

The 2 °C scenario (2DS) has been adopted in the Paris Agreement under the United Nations Framework Convention on Climate Change (UNFCCC), which requires limiting global warming to within 2 degrees Celsius above pre-industrial temperatures by the end of this century[1]. Accordingly, more and more Intended Nationally Determined Contributions (INDCs) spelling out post-2020 climate action have been submitted[2,3], where substantial effort must be made to reduce $CO_2$ emissions in energy-intensive sectors[4]. As one of the largest energy-intensive industries worldwide, the iron and steel industry is required to reduce $CO_2$ emissions by 50 Gt cumulatively through 2050 in the 2DS, contributing the largest share (35%) of $CO_2$ emission reductions among all industrial sectors[5]. However, driven mainly by the industrialisation in non-OECD countries[6], global crude steel production will grow >50% by 2050. As a consequence, the iron and steel industry is facing a severe challenge to accomplish the mid- to long-term decarbonisation target, under the scenario of an increasing $CO_2$ emission load but a diminishing $CO_2$ budget with time up to 2050 (Fig. 1).

Over the past decades, the iron and steel industry has reduced its energy consumption intensity by 60%, resulting in the current iron and steel production operating close to its thermodynamic

limits of ~20 gigajoules consumed per tonne of crude steel produced (Supplementary Figure 1). It is estimated that the room for further improvement in energy efficiency of the iron and steel industry is limited to 15–20%[7]. Therefore, improving the energy efficiency alone will not lead to the significant $CO_2$ emission reductions required by the 2DS, and carbon capture and storage (CCS) is the only current approach that could enable the sufficiently large reduction in $CO_2$ emissions required in this sector[8]. Conventional integrated steelmaking based on the blast furnace-basic oxygen furnace (BF-BOF) process (Fig. 2) is the current predominant production route, accounting for 74.2% of global crude steel production in 2015 and will continue to be a major approach for iron and steel production worldwide in the mid-to long-term[9]. In order to reduce the associated carbon intensity (Supplementary Figure 1), an increasing number of countries and regions have launched their own low-carbon steelmaking programmes, including Ultra-Low $CO_2$ Steelmaking (ULCOS)[10] in the European Union (EU) and $CO_2$ Ultimate Reduction in Steelmaking Process by Innovative Technology for Cool Earth 50 (Course50)[11,12] in Japan, in recent years. The current strategy of these programmes for decarbonisation of the iron and steel industry is to add on a separate $CO_2$ capture unit, based on various post-combustion $CO_2$ capture techniques, to the production facilities. The main attempts so far have been to retrofit the conventional iron and steel production facility to improve the performance of the $CO_2$ capture unit, including the top gas recycling-blast furnace, the HIsarna smelter, and the ULCORED and Corex processes[10]. To date, more than US$1 billion has been invested for these national and regional programmes[13]; nonetheless, development of the current decarbonisation options available for the iron and steel industry lags far behind the sectoral targets of $CO_2$ emission reduction. Still technical, economic, and political barriers need to be overcome to lower the yet costly decarbonisation expenses to an acceptable level, and options which are technically and economically ready for deployment in this sector are urgently needed.

In this study, the concept of steelmaking with inherent decarbonisation is proposed, based on the industrial CCS strategy, via making the best use of the limestone feedstock in a calcium-looping lime production (CaL-LP) scheme, which is demonstrated to be capable of achieving a substantial $CO_2$ emission reduction in a cost-effective manner. The techno-economic performance of the proposed concept for decarbonisation of the iron

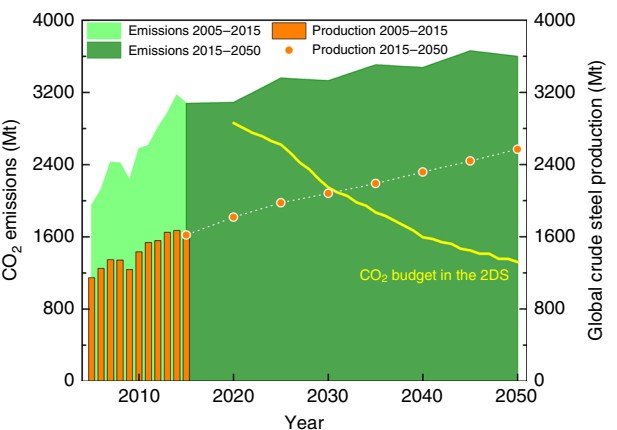

**Fig. 1** Sectoral contribution to $CO_2$ emission reduction in the 2 °C scenario. Evolution of annual crude steel production[9,44], $CO_2$ emissions[45,46], and the $CO_2$ budget[5] of global iron and steel industry up to 2050

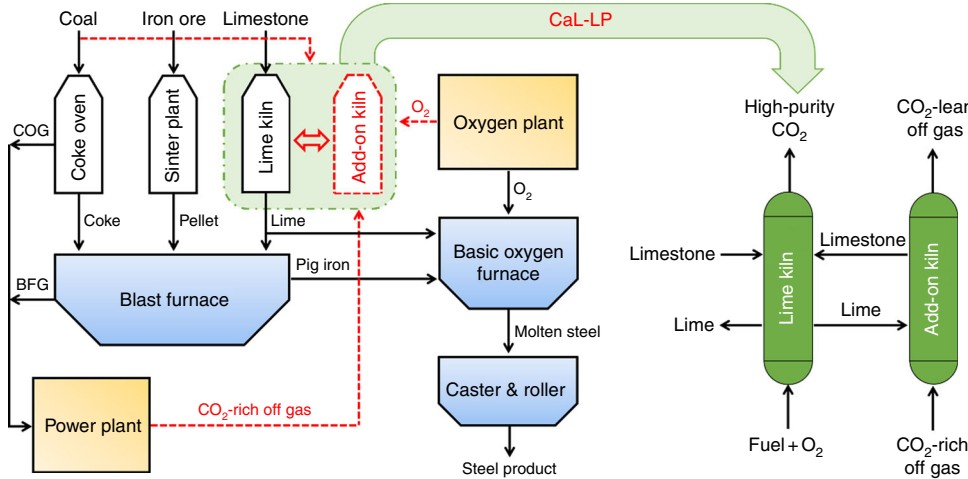

**Fig. 2** The concept of integrating decarbonisation into the iron and steel production. Schematic of the conventional integrated steel mill implementing the calcium-looping lime production (CaL-LP) scheme for steelmaking with inherent $CO_2$ emission reduction. The mass flows through the add-on kiln related to the CaL-LP scheme and existing manufacturing facilities in the steel mill are indicated with the red dashed and black solid lines, respectively. COG, coke oven gas; BFG, blast furnace gas

and steel industry is evaluated and compared with other currently considered decarbonisation options. Also, the potential of the CaL-LP scheme to meet the mid- to long-term decarbonisation target to 2050 is analysed, and the impact on steelmaking economics is assessed. Our results show that the significant $CO_2$ emission reduction resulting from the proposed CaL-LP scheme is enabled by avoiding conventional limestone calcination, process heat recovery, and direct $CO_2$ capture, which ensures that the steel mill will reach the 2DS decarbonisation target for 2050 by 2030. We find that the nature of the heat-powered process and minimised material requirement for the CaL-LP scheme result principally in a significant reduction in the $CO_2$ avoidance cost, which is less than one-third that of the benchmark amine scrubbing technology. We conclude that the techno-economic superiority places the CaL-LP scheme at the top of the available decarbonisation technologies for deployment in the global iron and steel industry, and that industrial decarbonisation is not necessarily expensive, as usually considered, as long as any CCS measure can be integrated deeply into the industrial manufacturing process.

## Results

**Concept process description.** The main manufacturing facilities in a conventional integrated steel mill can be classified into feedstock processing, iron and steel production, and ancillary units (Fig. 2). The feedstock processing unit (marked in white) includes the coke oven, sinter plant, and lime kiln, which pyrolyses coal into coke, sinters iron ore into pellets, and calcines limestone into lime, respectively. Coke reduces the pellets to pig iron in the blast furnace, where lime is fed to serve as a flux and to remove impurities from the produced pig iron. The final steel products are manufactured in the iron and steel production unit (light blue), including the blast furnace, basic oxygen furnace, caster, and roller. The power plant and oxygen plant are the main ancillary facilities (light orange), which provide power for all electricity-driven facilities in the mill and oxygen for steelmaking in the basic oxygen furnace, respectively.

Upon a relatively easy retrofitting of the lime kiln in this conventional integrated steelmaking process based on the calcium-looping $CO_2$ capture concept (light green)[14,15], the decarbonisation potential of the limestone feedstock can be exploited. Basically, as is depicted in Fig. 2, another kiln is added on to be interconnected with the lime kiln, and lime/limestone solids are circulated between the two kilns. In this way, $CO_2$ in the flue gas coming from the power plant is captured by lime delivered from the lime kiln via the exothermic carbonation reaction (Eq. 1) in the add-on kiln, and the limestone generated carries $CO_2$ from the add-on kiln to the lime kiln. In the lime kiln, lime is produced from oxy-fuel calcination (reverse Eq. 1) of the limestone circulated back from the add-on kiln and that freshly fed into the lime kiln, obtaining a high-purity $CO_2$ stream which is ready for storage or utilisation after compression. Meanwhile, a required portion of the lime produced is fed to the blast furnace and basic oxygen furnace, and the rest is circulated to the add-on kiln again as regenerated sorbent for a new calcium looping $CO_2$ capture cycle. As a consequence, the lime produced in the lime kiln is exploited to capture $CO_2$ in the add-on kiln before being used as a flux for iron and steel production. Such an industrial decarbonisation strategy of integrating a $CO_2$ emission reduction process deeply into the industrial manufacturing of materials has been investigated for cement production with low $CO_2$ emissions. Similarly, the limestone feedstock is processed in a calcium-looping operation where the spent lime, after $CO_2$ capture, is used as a raw material to manufacture cement in the rotary kiln[16–18]. However, the additional advantage in integrating

calcium looping into iron and steel production, as compared to cement production, lies in less modifications required to the manufacturing process of industrial products and the ancillary facilities (power plant and oxygen plant) which are required for calcium looping while readily available in steel mills.

$$CaO\,(s) + CO_2\,(g) \rightleftharpoons CaCO_3\,(s), \Delta H_{298\,K} = -178.2\,kJ/mol \quad (1)$$

The process flow diagram of the CaL-LP scheme and process simulaton in the add-on kiln can be found in detail in Supplementary Figures 2 and 3, respectively. The key parameter determining the reliability of simulating the CaL-LP scheme is the ratio of the limestone feedstock ($F_{0-CaCO_3}$) to the $CO_2$ to be disposed of ($F_{CO_2}$) in Supplementary Figure 2. On average, 0.3 t of limestone is consumed and 1.8 t of $CO_2$ is emitted to produce 1 t of crude steel in a conventional integrated steel mill[13]. Therefore, in the process simulation, the amount of limestone fed into the CaL-LP system is assigned to be equal to that (0.3 t/t crude steel) fed into the lime kiln in the conventional integrated steelmaking process, i.e., the yield of lime is not influenced compared to conventional lime production in the steel mill. Here, in order to evaluate the overall inherent decarbonisation potential of the limestone feedstock, the total $CO_2$ emitted from the steel mill (1.8 t/t crude steel) is assumed to be disposed with the CaL-LP scheme.

In the CCS framework, decarbonisation of the iron and steel industry includes two technical steps, i.e., capture of the $CO_2$ emitted and the subsequent storage or utilisation of the $CO_2$ captured. Regardless of the $CO_2$ capture approaches implemented, $CO_2$ storage is deployed following a generally accepted route, i.e., transporting $CO_2$ to the sequestration site via pressurised pipelines or shipping (longer distances over 1500 km), followed by subsurface storage for permanent $CO_2$ sequestration[19]. Actually, compared with $CO_2$ capture, $CO_2$ storage has already been technically proven through industrial projects of dedicated geological storage and $CO_2$-enhanced oil recovery ($CO_2$-EOR) at a scale over 1 Mt per year[20]. After decades of practice in the petroleum industry, $CO_2$-EOR has been a well-established technique which is economically favourable to enhance the recovery of oil from its depleted reservoirs through $CO_2$ injection[21,22]. EOR accounts for more than 90% of current industrial-scale projects for $CO_2$ storage[19]; and importantly, it has been estimated that global $CO_2$ storage capacity of oil fields approaches 350 Gt[22], which is more than sufficient to cover the cumulative $CO_2$ emission reductions (50 Gt) from the global iron and steel industry required in the 2DS. Therefore, the techno-economic feasibility for the successful deployment of any decarbonisation option in this industrial sector depends highly on the $CO_2$ capture step, which is the main focus in the discussions below.

**Techno-economic feasibility.** Fig. 3 reveals the decarbonisation potential of an integrated steel mill implementing the CaL-LP scheme and evolution of the resultant $CO_2$ emission intensity up to 2050. Depending on the operating parameters employed, 49.0–83.8% of the total $CO_2$ emissions in an integrated steel mill can be reduced inherently during lime production. Under extreme operating conditions (lower limestone feed percent), the scheme is capable of capturing almost all the $CO_2$ emitted. The total $CO_2$ emission reduction resulting from the CaL-LP scheme is attributed to the decrease in direct $CO_2$ emissions due to avoidance of the conventional limestone calcination process and partial recovery of the energy consumed in the lime kiln, and capture of the $CO_2$ emitted due to lime carbonation in the add-on kiln. Encouragingly, the resulting overall $CO_2$ emission reduction

surpassed the technical target put forward by Japan and the EU, which aim to reduce $CO_2$ emissions in the iron and steel industry by ~30% through the COURSE50 programme and at least 50% through the ULCOS programme, respectively. According to the 2DS, total allowable $CO_2$ emissions to 2050 are limited to 61 Gt for the global iron and steel industry, which requires the annual $CO_2$ emission from this sector to be reduced constantly from current levels of 3.1 Gt to 1.3 Gt by 2050 (Fig. 1). On the other hand, the annual production of crude steel worldwide is projected to keep increasing to more than 2.5 Gt per year in 2050. Therefore, the average $CO_2$ emission intensity of the global iron and steel industry has to be reduced stepwise from 1.57 t/t crude steel in 2020 to 0.51 t/t crude steel in 2050. Regardless of the limestone feed percent employed, the CaL-LP scheme ensures that $CO_2$ emission intensity of the steel mill is reduced sufficiently to meet the target values in the 2DS to 2030. More importantly, when calcium-looping production of lime is operated at a limestone feed percent below 4%, the 2DS $CO_2$ emission target for 2050 can be reached by 2030.

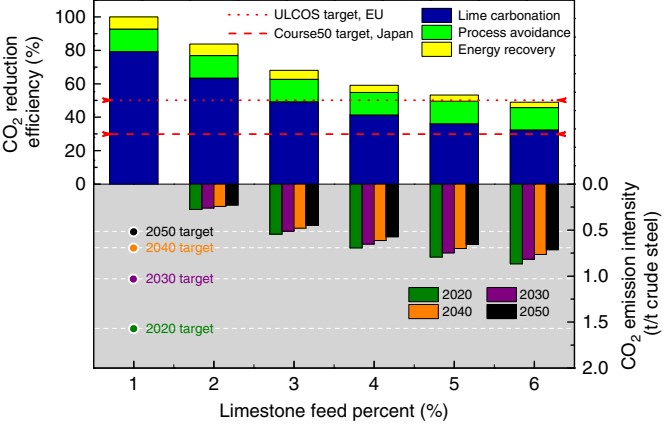

**Fig. 3** Decarbonisation potential of a conventional integrated steel mill implementing the CaL-LP scheme. The $CO_2$ reduction efficiency resulting from lime carbonation, process avoidance, and energy recovery is shown as a function of the limestone feed percent, and compared with technical targets of the COURSE50 and ULCOS programmes; The $CO_2$ emission intensity from 2020 to 2050 is shown as a function of the limestone feed percent, and compared with the sectoral decarbonisation targets through 2050 in the 2DS

The techno-economics of various decarbonisation options currently considered for the iron and steel industry[23–27] are compared with that of the CaL-LP scheme in Fig. 4a in terms of the primary energy consumption and $CO_2$ avoidance cost. Amine scrubbing (AS)[28,29], pressure swing adsorption (PSA)[30], and membrane separation (MS)[31,32] are the three most widely investigated $CO_2$ capture technologies for the iron and steel industry, regardless of the type of steelmaking process. As a current benchmark industrial $CO_2$ capture technology, AS can achieve a primary energy consumption for $CO_2$ emission reduction of ~3.5 GJ/t when applied in the iron and steel industry, with the $CO_2$ avoidance cost ranging between 45–60 €$_{2010}$/t. In comparison with AS, the MS technology has the potential to minimise the primary energy consumption for $CO_2$ emission reduction to below 2.0 GJ/t, while the PSA technology has the potential to minimise the $CO_2$ avoidance cost to below 30 €$_{2010}$/t.

With regard to the CaL-LP scheme, the resulting primary energy consumption for $CO_2$ emission reduction can be reduced to approach the situation when applying the MS and PSA technology. Importantly, the CaL-LP scheme significantly reduces the $CO_2$ avoidance cost, when compared with other decarbonisation options, which is almost half that of PSA and less than one-third that of AS. It is worth mentioning that, among all decarbonisation options included in Fig. 4a, the CaL-LP scheme is the only option exhibiting a $CO_2$ avoidance cost lower than the $CO_2$ price under the EU Emission Trading System[33] in 2020, whilst other options will not become economically feasible until 2030, considering the projected $CO_2$ price. According to the breakdown of $CO_2$ avoidance costs due to implementation of the CaL-LP scheme or other considered decarbonisation options[27,34] in Fig. 4b and Supplementary Figure 4, the significantly lower energy cost of the CaL-LP scheme than those of other decarbonisation options results principally in the decline in the $CO_2$ avoidance cost, albeit a limited difference in the primary energy consumption for $CO_2$ emission reduction between the CaL-LP scheme and other options. This is attributed to the nature of a heat-powered process for the CaL-LP scheme, rather than other decarbonisation options which are powered completely by electricity.

The breakdown of the incremental primary energy consumption due to $CO_2$ emission reduction in Fig. 5 reveals that separation of $O_2$ is the only electricity-powered step in the CaL-LP scheme except the compression of $CO_2$, which is a required step for all the mentioned decarbonisation options, and those

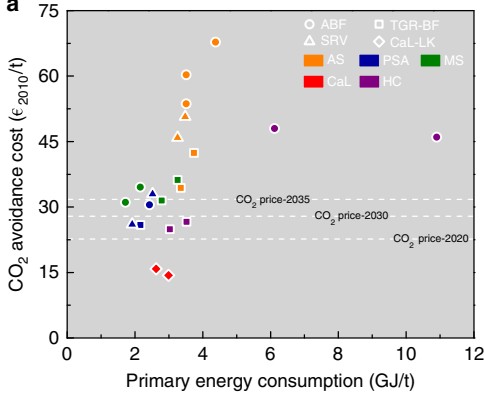

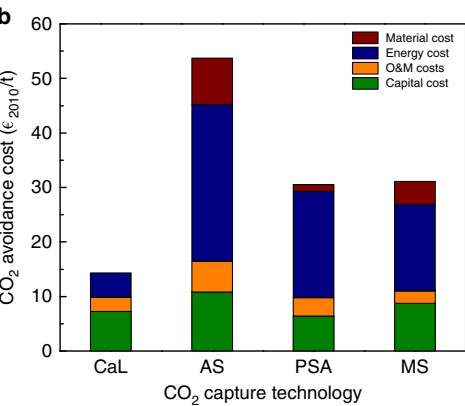

**Fig. 4** Techno-economic superiority of the CaL-LP scheme for low-carbon steelmaking. **a** $CO_2$ avoidance cost and primary energy consumption, and **b** cost structure of various options to reduce $CO_2$ emissions in the iron and steel industry. The iron and steel production facilities: ABF, air-blown blast furnace; TGR-BF, top gas recycling-blast furnace; SRV, smelting reduction vessel; CaL-LK, calcium-looping lime kiln; and the $CO_2$ capture technologies: AS, amine scrubbing; PSA, pressure swing adsorption; MS, membrane separation; CaL, calcium looping; HC, hydrate crystallisation

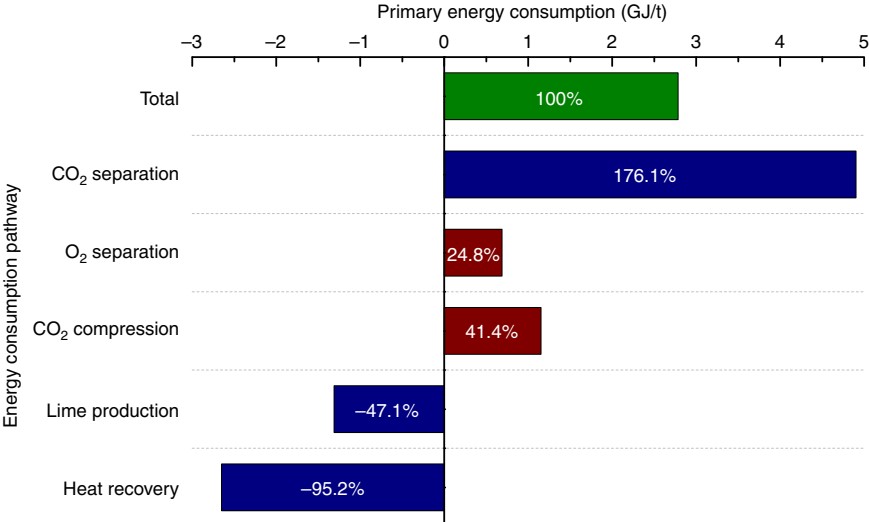

**Fig. 5** Energy consumption characteristics of the CaL-LP scheme. Breakdown of the primary energy consumption for $CO_2$ emission reduction in an integrated steel mill implementing the CaL-LP scheme at a typical limestone feed percent of 5%. Green bar - the total incremental primary energy consumption; blue bar - heat-powered energy consumption steps; and red bar - electricity-powered energy consumption steps

heat-powered steps (blue bar) dominate the total primary energy consumption for $CO_2$ emission reduction. Hence, primary energy in the fuel can be exploited directly, accompanied with the process heat recovery[35] (see Supplementary Figure 5), instead of being transformed into electricity before driving any $CO_2$ capture processes. In consequence, the energy cost is decreased appreciably[36] and the electricity price is no longer a factor hindering deployment of such a decarbonisation option[25]. In addition, on account of integrating decarbonisation into the steelmaking process via making full use of the limestone feedstock in the CaL-LP scheme (Fig. 2), the material cost of $CO_2$ avoidance is minimised to almost zero. This is another favourable nature superior to other decarbonisation options for which a high proportion of the total $CO_2$ avoidance cost goes toward materials for $CO_2$ separation, such as amine for AS, zeolite for PSA, and membrane for MS. Therefore, the economic superiority considering $CO_2$ prices and other current decarbonisation options would ensure that the CaL-LP scheme is a promising candidate competing for the best available technology for decarbonisation of the global iron and steel industry in the mid- to long-term timeframe.

**Impact on steelmaking economics**. The impact on steelmaking is a key factor determining the feasibility for commercial implementation of any decarbonisation measure in the iron and steel industry. The incremental cost of steel, indicative of the economic impact, resulting from the implementation of various decarbonisation options is shown in Fig. 6, with the corresponding $CO_2$ emission intensity included for comparison. As can be seen, implementation of $CO_2$ capture techniques in the air-blown blast furnace and smelting reduction vessel—conventional iron and steel production facilities—could provide a low and currently acceptable incremental cost of steel (~20 €$_{2010}$/t); however, the resulting $CO_2$ emission intensity of ~1.5 t/t crude steel makes it difficult to reach the 2DS decarbonisation target for the global iron and steel industry from 2020 onward (Fig. 3). An appreciably lower $CO_2$ emission intensity can be achieved through applying $CO_2$ capture techniques into the innovative steelmaking facilities, such as the top gas recycling-blast furnace and water-gas shifted furnace; but this will come at the cost of a substantial increase in the incremental cost of steel to more than 40 €$_{2010}$/t. This increase is mainly attributed to the additional capital requirement due to

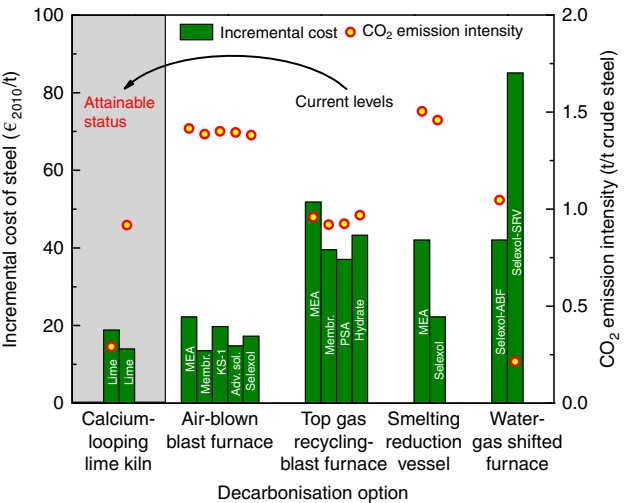

**Fig. 6** Incremental cost of steel and the corresponding $CO_2$ emission intensity of the steel mill implementing various decarbonisation options. MEA, monoethanolamine[47]; KS-1, KS-1$^{TM}$ solvent[48]; Adv. sol., advanced solvents[49]; Selexol, Selexol$^{TM}$ physical absorption process[50]; Membr., membrane separation using selective carbon membrane[51]; Hydrate, hydrate crystallisation $CO_2$ capture process[24]

retrofitting of the conventional iron and steel production facilities, and energy consumption associated with the increased decarbonisation levels. Importantly, when the same level of $CO_2$ emission intensity is achieved, the resulting incremental cost of steel due to the CaL-LP scheme dropped to less than half that in the case of other decarbonisation options. A few options exhibit a low incremental cost of steel comparable with that of the CaL-LP scheme; however, the corresponding $CO_2$ emission intensity does not decline as much as for the CaL-LP scheme. Hence, it can be concluded that the CaL-LP scheme provides the optimal economics for steelmaking with low $CO_2$ emissions among all current decarbonisation options, since it enables a drastic decrease in both the $CO_2$ emission intensity and incremental cost of steel simultaneously. Overall, the incremental cost of steel can be controlled to as low as 3.2–5.3% of the total manufacturing cost of steel[37] using the CaL-LP scheme for $CO_2$ emission reduction (see

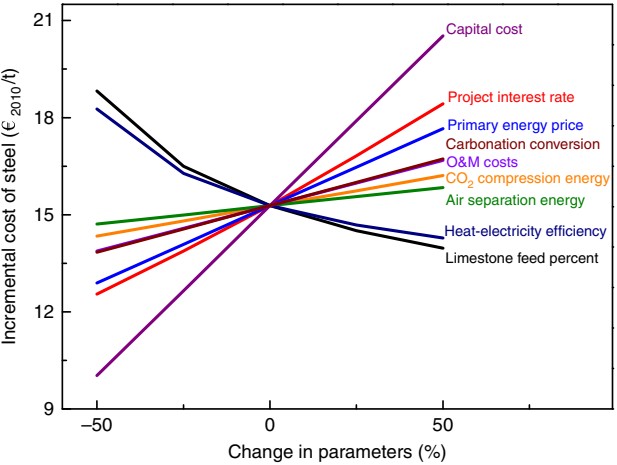

**Fig. 7** Sensitivity of the incremental cost of steel due to implementation of the CaL-LP scheme. The influence of nine potentially key parameters on the incremental cost of steel was investigated by changing the value of each parameter up to ± 50%, while holding all other parameters at their base values

Supplementary Figure 6). However, it should be pointed out that, in addition to the inevitable incremental costs, any implemented decarbonisation option may influence the properties of iron and steel products, which should be explored before their deployment.

In addition, we analysed the sensitivity of nine potentially key parameters during operation of the CaL-LP scheme on the steelmaking economics. As is depicted in Fig. 7, all lines intersect at the same incremental cost of steel of 15.28 €$_{2010}$/t, when all parameters remain unchanged from their base values. Each parameter, once varied, will lead to the change of resulting incremental costs of steel following a different slope. Among all lines, the one resulting from the change in capital cost showed the steepest slope, indicating that capital cost has the greatest effect on the incremental cost of steel. This is easy to understand when we notice that capital cost dominates the cost structure of $CO_2$ avoidance with the CaL-LP scheme (Fig. 4b). However, the resulting incremental cost of steel ranges within 10.03–20.53 €$_{2010}$/t, less than 35% change from the base case value. Other notable parameters in terms of the sensitivity include project interest rate, primary energy price, limestone feed percent, and heat-electricity efficiency. Primary energy price is unlikely to decrease in future unless fossil fuels can be replaced by cheaper renewable energy, while adjusting the project interest rate provides an alternative way to lower the incremental cost of steel. Limestone feed percent and heat-electricity efficiency show a negative correlation with the incremental cost of steel and are more influential in increasing than in decreasing the incremental cost of steel. The remaining four analysed parameters, i.e., operation and maintenance (O&M) costs, average carbonation conversion of lime, air separation energy, and $CO_2$ compression energy, are revealed to have much lower impacts on the economics of steelmaking with the CaL-LP scheme. In particular, the increase in the average carbonation conversion, indicating the reactivity of lime to $CO_2$, will lead to a significant improvement in the decarbonisation efficiency (Supplementary Figure 7) within a limited increase in the incremental cost of steel. Technically, steam hydration of lime is a promising choice to enhance its reactivity[38,39], while the economic feasibility to add such a step into the proposed CaL-LP scheme requires further evaluation.

## Methods
**Process thermodynamic simulation**. According to the process flow diagram depicted in Supplementary Figure 2, the maximum average carbonation conversion

($X_{ave,max}$) of lime in the add-on kiln is a key process parameter to evaluate the performance of the CaL-LP scheme. Here, the method proposed by Abanades[40], which has been widely used to determine the $CO_2$ capture capacity of calcium-looping sorbents, is employed to calculate $X_{ave,max}$ via equation 2. In this, the value of $X_{ave,max}$ is directly related to the mass fraction ($r_N$) and carbonation conversion ($X_N$) of the lime particles that have been circulated $N$ times through the looping scheme in Supplementary Figure 2, and determination of the variables $r_N$ and $X_N$ can be found in detail in Supplementary Note 1 and Supplementary Table 1. In fact, the lime carbonation conversion in actual operation of the CaL-LP scheme will be affected by material attrition and fragmentation under fluidisation conditions, lime sulphation, and ash accumulation. These issues are of a complexity that few of the currently developed models could effectively deal with[15], and thus, are not considered in this study.

$$X_{ave,max} = \sum_{N=1}^{\infty} r_N \cdot X_N. \tag{2}$$

When $X_{ave,max}$ is determined, the energy balance in the lime kiln of the CaL-LP scheme can be described as:

$$E_{CO_2\,separation} = E_{cycled\,solid} + E_{fresh\,solid} + E_{O_2} + E_{reaction}, \tag{3}$$

where the total primary energy input into the lime kiln ($E_{CO_2\,separation}$), also indicative of the energy required to separate $CO_2$ in the kiln, is consumed through four pathways. They are: heating the solids circulated from the add-on kiln ($E_{cycled\,solid}$); the fresh limestone ($E_{fresh\,solid}$) and oxygen ($E_{O_2}$) fed into the lime kiln up to the calcination temperature; and driving the endothermic limestone calcination reaction ($E_{reaction}$), respectively. Determination of each term in Eq. 3 can be found in detail in Supplementary Note 2 and Supplementary Tables 2 and 3.

The overall energy balance of the conventional integrated steel mill retrofitted with the CaL-LP scheme can be described as:

$$\Delta E_{fuel} = E_{CO_2\,separation} + E_{O_2\,separation} + \\ E_{CO_2\,compression} - E_{lime\,kiln} - E_{heat\,recovery}, \tag{4}$$

where the total incremental primary energy consumption due to implementation of the process retrofitting scheme ($\Delta E_{fuel}$) is directly related to the energy required to separate $CO_2$ in the lime kiln ($E_{CO_2\,separation}$) and $O_2$ in the onsite oxygen plant ($E_{O_2\,separation}$), and to compress $CO_2$ ($E_{CO_2\,compression}$). However, these energy consumptions can be offset partially by saving the energy consumed for lime production in the conventional lime kiln ($E_{lime\,kiln}$) and recovering the process heat released due to the exothermic carbonation reaction of lime in the add-on kiln ($E_{heat\,recovery}$)[35]. Determination of each term in Eq. 4 can be found in detail in Supplementary Note 3 and Supplementary Tables 2–4.

**Decarbonisation evaluation**. Carbon dioxide balance of the conventional integrated steel mill retrofitted with the CaL-LP scheme can be analysed via Eq. 5, where the $CO_2$ generation due to oxygen separation ($M_{CO_2}^{O_2\,separation}$) and $CO_2$ compression ($M_{CO_2}^{CO_2\,compression}$), and the $CO_2$ avoidance from conventional lime production ($M_{CO_2}^{lime\,kiln}$) and carbonation heat recovery ($M_{CO_2}^{heat\,recovery}$) result in the difference in direct $CO_2$ emissions between the reference mill ($M_{CO_2}^{reference}$) and the retrofitted one ($M_{CO_2}^{retrofit}$). Determination of related terms in Eq. 5 can be found in detail in Supplementary Note 4.

$$M_{CO_2}^{retrofit} = M_{CO_2}^{reference} + M_{CO_2}^{O_2\,separation} + \\ M_{CO_2}^{CO_2\,compression} - M_{CO_2}^{lime\,kiln} - M_{CO_2}^{heat\,recovery}. \tag{5}$$

Therefore, overall decarbonisation potential ($\varphi_{CO_2}$) of the conventional integrated steel mill implementing the CaL-LP scheme is calculated according to Eq. 6, once the $CO_2$ capture efficiency ($\eta_{capt.}$) in the add-on kiln is determined.

$$\varphi_{CO_2} = \frac{M_{CO_2}^{reference} - \left(1 - \eta_{capt.}\right) \cdot M_{CO_2}^{retrofit}}{M_{CO_2}^{reference}} \times 100\%. \tag{6}$$

**Cost structure analysis**. We employ $CO_2$ avoidance cost (AC, €/t), a standardised cost measure of $CO_2$ capture defined by IPCC (Eq. 7)[41], to assess the economic performance of the CaL-LP scheme proposed for decarbonisation of the iron and steel industry,

$$AC = \frac{(COS)_{retrofit} - (COS)_{reference}}{(EOC)_{reference} - (EOC)_{retrofit}}, \tag{7}$$

where COS represents the levelised cost of steel, and EOC represents the emission of $CO_2$ per tonne of steel produced. The subscripts retrofit and reference refer to

**Table 1 Main techno-economic assumptions employed in this study**

| Parameter | Unit | Value |
|---|---|---|
| Carbonation temperature (add-on kiln) | K | 923.15[15] |
| Calcination temperature (lime kiln) | K | 1173.15[15] |
| Lime carbonation enthalpy | kJ/mol | −171.6[52] |
| Limestone calcination enthalpy | kJ/mol | 166.0[52] |
| Fuel heat value | MJ/kg | 29.3[53] |
| Carbon-to-calcium ratio[a] | kg/kg | 6.0[13] |
| Economic lifetime | yr | 25[47] |
| Project interest rate | % | 10[42] |
| Primary energy price | €$_{2010}$/GJ | 1.5[33] |
| $CO_2$ compression energy | kWh/t | 110.9[54] |
| Air separation energy | kWh/t $O_2$ | 184.8[55] |
| Capital cost | €$_{2010}$/t crude steel | 70[56] |
| O&M costs | %-capital cost | 4[57] |

[a]The mass ratio of the total $CO_2$ emitted to the limestone feedstock consumed in a conventional integrated steel mill

the steel mill with and without the CaL-LP scheme, respectively. Upon introduction of the method calculating manufacturing cost of industrial products as proposed by Kuramochi et al.[42] (Supplementary Note 5), the specific $CO_2$ avoidance cost in the iron and steel industry can be determined via Eq. 8,

$$AC = \frac{\alpha \cdot \Delta I' + \Delta C'_{Energy} + \Delta C'_{O\&M} + \Delta C'_{Material}}{(EOC)_{reference} - (EOC)_{retrofit}}, \qquad (8)$$

where $\Delta I'$, $\Delta C'_{Energy}$, $\Delta C'_{O\&M}$, and $\Delta C'_{Material}$ represent the additional capital requirement, annual cost of energy, annual operation and maintenance (O&M) costs, and annual cost of raw materials per tonne of steel produced, respectively, due to implementation of the CaL-LP scheme.

The key techno-economic assumptions employed for process thermodynamic simulation and cost structure analysis are presented in Table 1. Integration of data, models, and methods involved in this study for result acquisition is depicted in Supplementary Figure 8 and Supplementary Note 6. In order to compare the cost figures drawn in this study with literature values, we apply the Chemical Engineering Plant Cost Index (CEPCI)[43] to eliminate the influence of labour and price on the cost figure in different periods, and convert all cost figures to €$_{2010}$.

## Data availability

The authors declare that the data supporting the findings of this study are available within the article and its Supplementary Information file, and from the corresponding author upon reasonable request.

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

## Acknowledgements

We gratefully acknowledge the 2017 Macquarie University Research Fellowship (MQRF) Scheme for S.T. and National Natural Science Foundation of China (Grant No. 21776160) for financial support. National Natural Science Fund for Outstanding Young Scholars of China (Grant No. 51522401) and Engineering and Physical Sciences Research Council (EPSRC, Grant No. EP/P034594/1) are also acknowledged.

## Author contributions

S.T. and J.J. conceived the study. S.T. and Z.Z. collected the data. S.T. built the research methodology and performed all calculations. S.T. and V.M. conducted the analysis. S.T. wrote the paper. All authors discussed the results and commented on the manuscript.

## Additional information

**Competing interests:** The authors declare no competing interests.

