## [Peer Review File · Nature Communications]

Reviewers' comments:

Reviewer #1 (Remarks to the Author):

The authors have presented their work on evaluating the potential for steelmaking with inherent decarbonisation using limestone feedstock in a calcium looping lime production (Cal-LP) scheme. The authors provide a perspective on the potential for using calcium looping to meet the decarbonisation targets set up by the Paris Agreement. While the work is of value to researchers in the field, the manuscript lacks in the following technical aspects of Calcium based work.

1. The success of any decarbonisation scheme is inherently dependent on the efficacy of carbon capture and the efficacy of sequestration. While the manuscript is focused on carbon capture aspect of the decarbonisation target, it is impossible to get a holistic conclusion without considering the sequestration aspect of decarbonization.

2. Cal-LP studies cited in the literature and cited in this study assume an optimistic value for the calcium to carbon ratio and its continuous circulation. The Calcium to carbon ratio is an important parameter, which determines the feasibility of a Cal-LP system and is missing from this study.

3. A major technical challenge for the working of a Calcium looping system is the loss of reactivity of sorbent as a function of time. The sensitivity study does not include this parameter to provide a realistic assessment for the readers of this article.

4. Over past several years, a hydration step for Calcium looping system has shown to have several advantages over a two reactor calcium looping system, wherein the inclusion of hydration as a step is shown to ensure that the calcium-based loop operates at near stoichiometric ratio. These three reactor systems have shown advantages over the conventional two reactor Calcium looping systems. The authors do not include a justification for the use of a two-reactor system as opposed to a three-reactor system. Examples of the hydration based calcium looping system can be found in articles like: "Coppola, Antonio, et al. "Reactivation by steam hydration of sorbents for fluidized-bed calcium looping." *Energy & Fuels* 29.7 (2015): 4436-4446."

Reviewer #2 (Remarks to the Author):

Review of the Ms.:

Inherent potential of global iron and steel industry to exceed the mid- to long-term decarbonisation targets

by S. Tian et al.

This is a work focusing on a topic of interest for the Readers of this Journal, as it deals with the concept of integrating iron and steel production with a decarbonisation process – calcium looping. Generally speaking, the article has been well structured and written. Nonetheless, some relevant issue should still be solved – at least in the Reviewer's opinion – before reconsidering it for final publication. Details are given below.

1. The general framework in which the concept has been developed is that concerning coupling between industrial production of materials and carbon dioxide capture and concentrated release with contextual use of lime. Other examples have been given in literature, such as those concerning the integration of cement production with calcium looping. Authors could make, in the Introduction, a comparison between their concept and the one referring to cement production, highlighting similarities and differences. This would give to an interested Reader a more complete

scenario on the subject.

2. The supplementary information embodies data and concepts that, at least in part, would be better to read in the main article. If the word count requirements for this Journal allow this, the Authors should re-equilibrate the two files by letting some of the relevant information contained in the supplementary file migrate into the main file (which, at least in general, is the most read between the two).

3. The Reader could miss a synoptic Table referring to the main operating conditions of the whole system. Information has been given in distributed way, but much more comfortable would be a direct Table containing the most important parameters. It should be given in the main article file.

4. It seems that aspects concerning sorbent size population have not been profoundly taken into account. How would the following issues be interrelated, with specific reference to the calcium looping sub-process? 1) General hypotheses of the model; 2) Fluidisation conditions; 3) Sorbent particle physico-chemical properties; 4) Extent of attrition and fragmentation phenomena; 5) Population balance on sorbent particle size; 6) Changes in the residence time distribution for sorbent particles following the changes in their particle size distribution; 7) Carbon capture performance following the changes in particle size and residence time distributions; 8) Fluxes of solid and gas exiting the calcium looping apparatus; 9) Integration with the steel production process.

Reviewer #3 (Remarks to the editor only):

Reviewer #3 left comments for the editor only. They remark that such a strong CO₂ emission reduction (>68%) seems improbable given that in primary steel making lime consumption represents half that of coal consumption and they seek clarification on this matter. As this point is of the most interest for our journal, substantial support to this conclusion is required. Furthermore, they note that there are missing chemical formulas regarding the explanation of the carbon looping process, and they feel figures 3-6 could benefit from improved explanations.

Response to Reviewers

Reviewer #1 (Remarks to the Author):

The authors have presented their work on evaluating the potential for steelmaking with inherent decarbonisation using limestone feedstock in a calcium looping lime production (CaL-LP) scheme. The authors provide a perspective on the potential for using calcium looping to meet the decarbonisation targets set up by the Paris Agreement. While the work is of value to researchers in the field, the manuscript lacks in the following technical aspects of Calcium based work.

1. The success of any decarbonisation scheme is inherently dependent on the efficacy of carbon capture and the efficacy of sequestration. While the manuscript is focused on carbon capture aspect of the decarbonisation target, it is impossible to get a holistic conclusion without considering the sequestration aspect of decarbonization.

Author's response: Thank you for your advice. We fully agree that it will be of great importance to include discussions about the sequestration aspect, which will make this study succeed in conveying a holistic picture to the audience on the concept of decarbonisation. We have added a passage to explain our considerations of the sequestration aspect of this study in Lines 142–159 (Page 6) of the revised manuscript. Briefly speaking, the sequestration aspect is currently less technically challenging as compared to the carbon capture aspect, and the techno-economic feasibility for the successful deployment of the CaL-LP scheme for decarbonisation of the iron and steel industry depends highly on the carbon capture aspect.

2. CaL-LP studies cited in the literature and cited in this study assume an optimistic value for the calcium to carbon ratio and its continuous circulation. The Calcium to carbon ratio is an important parameter, which determines the feasibility of a CaL-LP system and is missing from this study.

Author's response: Thanks for your comments. Actually, we have discussed the “calcium-to-carbon ratio” at the end of the “Concept process description” sub-section in the original manuscript. We are fully aware that the “calcium-to-carbon ratio”, i.e., the ratio of the limestone

feedstock to the CO₂ to be disposed of, is the key parameter determining the reliability of simulating the CaL-LP scheme for decarbonisation of the iron and steel industry. Here in this study, for the sake of CO₂ emission reduction from the iron and steel industry, we have used the ratio of 0.3 t of limestone to 1.8 t of CO₂ according to the practical iron and steel production process (World Steel Association, <https://www.worldsteel.org/en/dam/jcr:5b246502-df29-4d8b-92bb-afb2dc27ed4f/Sustainable-steel-at-the-core-of-a-green-economy.pdf>), rather than those ratios from the literatures usually related to calcium looping CO₂ capture from coal-fired power plants or cement plants. In addition, the average carbonation conversion of lime is another key parameter influencing the CO₂ emission reduction efficiency of calcium looping. For fear of overestimating the decarbonisation potential of the iron and steel industry through the proposed CaL-LP scheme, we compared different semi-empirical models commonly considered for characterisation of the carbonation conversion of CaO-based CO₂ sorbents (Supplementary Table 1), and employed a relatively conservative model to ensure a reliable prediction in this study. Here in the revised manuscript, we have more clearly explained the “calcium-to-carbon ratio” in Lines 130–141 (Page 6), and now believe that the issue has been properly clarified.

3. A major technical challenge for the working of a Calcium looping system is the loss of reactivity of sorbent as a function of time. The sensitivity study does not include this parameter to provide a realistic assessment for the readers of this article.

Author’s response: Thanks for your comments. Actually, we have included the parameter indicating the loss of reactivity of sorbent in the sensitivity analysis, which is the “average carbonation conversion of lime” in Figure 6 of the original manuscript (Figure 7 in the revised manuscript). The loss in the sorbent reactivity will directly lead to the decrease in the average carbonation conversion of the sorbent, as will the reverse case. Here in the revised manuscript, we have added discussions in Lines 323–326 (Page 15) to provide a realistic assessment of this parameter as required by the reviewer.

4. Over past several years, a hydration step for Calcium looping system has shown to have several advantages over a two reactor calcium looping system, wherein the inclusion of hydration as a step is shown to ensure that the calcium-based loop operates at near stoichiometric ratio. These three reactor systems have shown advantages over the conventional two reactor

Calcium looping systems. The authors do not include a justification for the use of a two-reactor system as opposed to a three-reactor system. Examples of the hydration based calcium looping system can be found in articles like: “Coppola, Antonio, et al. "Reactivation by steam hydration of sorbents for fluidized-bed calcium looping." *Energy & Fuels* 29.7 (2015): 4436-4446.”

Author's response: Thanks for your advice. The hydration-enhanced calcium looping process, i.e., the three-reactor system the reviewer has mentioned, has indeed been recently demonstrated to be technically superior to the conventional two-reactor calcium looping process referred to in this study. However, the conventional calcium looping process has experienced advanced development in lab exploration, pilot-plant testing, and process modelling as compared to the emerging hydration-enhanced process. Considering that this study is a modelling work aiming at demonstrating the techno-economic superiority of integrating a calcium-looping lime production scheme for steelmaking with low CO₂ emissions over all other decarbonisation options currently considered for the iron and steel industry, we prefer to employ the conventional two-reactor calcium looping process. This is because more techno-economic assumptions and operating parameters, which are reliable for the modelling work, are related to the conventional two-reactor process. As for the mentioned hydration-included three-reactor process, we agree with the reviewer that steam hydration is a promising choice to appreciably improve the performance of CaO-based CO₂ sorbents in calcium looping and, thus, deserves to be discussed in this study. Hence, we have added discussions in Lines 326–329 (Page 15) of the revised manuscript to introduce this promising technique to the readers, who can find details in the references we have cited.

Reviewer #2 (Remarks to the Author):

This is a work focusing on a topic of interest for the Readers of this Journal, as it deals with the concept of integrating iron and steel production with a decarbonisation process – calcium looping. Generally speaking, the article has been well structured and written. Nonetheless, some relevant issue should still be solved – at least in the Reviewer's opinion – before reconsidering it for final publication. Details are given below.

1. The general framework in which the concept has been developed is that concerning coupling between industrial production of materials and carbon dioxide capture and concentrated release with contextual use of lime. Other examples have been given in literature, such as those concerning the integration of cement production with calcium looping. Authors could make, in the Introduction, a comparison between their concept and the one referring to cement production, highlighting similarities and differences. This would give to an interested Reader a more complete scenario on the subject.

Author's response: Thanks for your advice. It is really a good idea to make a comparison between the proposed CaL-LP scheme in this study and recent studies regarding CaL-integrated cement production. We believe it is more beneficial for the readers to provide this comparison right after the process description of the CaL-LP scheme than in the Introduction. Therefore, in Lines 118–127 (Page 5) of the revised manuscript we have added text to briefly introduce recent progress in the research on CaL-integrated cement production, highlighting similarities and differences as compared to the CaL-LP scheme.

2. The supplementary information embodies data and concepts that, at least in part, would be better to read in the main article. If the word count requirements for this Journal allow this, the Authors should re-equilibrate the two files by letting some of the relevant information contained in the supplementary file migrate into the main file (which, at least in general, is the most read between the two).

Author's response: Thanks for your advice. We have carefully gone through both files again, and feel that it would really make the manuscript more readable to move Supplementary Fig. 5 and the main techno-economic assumptions in Supplementary Tables 2 and 4 from the Supplementary Information file to the main file, which are now Figure 5 in Lines 230–236 (Page 11) and Table 1 in Lines 431–433 (Page 19), respectively, in the revised manuscript. In addition, we have also added text, in the revised manuscript, in Lines 238–242 (Page 11) to discuss the results in Figure 5 and in Lines 435–436 (Page 20) to introduce Table 1.

3. The Reader could miss a synoptic Table referring to the main operating conditions of the whole system. Information has been given in distributed way, but much more comfortable would

be a direct Table containing the most important parameters. It should be given in the main article file.

Author's response: Thanks for your advice. We have introduced a synoptic Table summarising the main operating conditions/parameters of the proposed scheme as Table 1 in Lines 431–433 (Page 19) in the revised manuscript.

4. It seems that aspects concerning sorbent size population have not been profoundly taken into account. How would the following issues be interrelated, with specific reference to the calcium looping sub-process? 1) General hypotheses of the model; 2) Fluidisation conditions; 3) Sorbent particle physico-chemical properties; 4) Extent of attrition and fragmentation phenomena; 5) Population balance on sorbent particle size; 6) Changes in the residence time distribution for sorbent particles following the changes in their particle size distribution; 7) Carbon capture performance following the changes in particle size and residence time distributions; 8) Fluxes of solid and gas exiting the calcium looping apparatus; 9) Integration with the steel production process.

Author's response: Thanks very much for your valuable advice related to a further understanding of the CaL-LP scheme in terms of the sorbent size population. The 9 points you listed here outline an integral logic to study this aspect, which is a technical issue of great importance to be investigated when we move this research forward to practical application. This study is a modelling work on the techno-economic feasibility of the proposed CaL-LP scheme for decarbonisation of the iron and steel industry in the mid- to long-term timeframe, which is based on the integration of thermodynamic and economic models as depicted in Supplementary Fig. 8. We agree that the change in the size distribution of sorbent particles, which is mainly due to material attrition and fragmentation, can affect the CO₂ capture efficiency of the calcium looping sub-process. This aspect primarily belongs to the scope of reaction hydrodynamics in both kilns. In fact, there have been several semi-predictive models with simple hydrodynamics or predictive models with CFD hydrodynamics developed for a more accurate simulation of the calcium looping process (Hanak et al., *Energy Environ. Sci.*, 2015, 8, 2199–2249), however, none of these models accounts for the change in the particle size distribution due to sorbent attrition and fragmentation. Most studies concerning this issue relied on experimental demonstrations (Erans et al., *Appl. Energy*, 2018, 225, 392–401; Erans et al., *Fuel*, 2017, 187,

388–397). Therefore, we believe that the integrated techno-economic model employed in this study is reliable at the current concept design stage in demonstrating the techno-economic superiority of the proposed steelmaking scheme with low CO₂ emissions. We have added text to explain our considerations of this issue in Lines 374–378 (Page 17) of the revised manuscript.

Reviewer #3 (Remarks to the editor only):

Reviewer #3 left comments for the editor only. They remark that such a strong CO₂ emission reduction (>68%) seems improbable given that in primary steel making lime consumption represents half that of coal consumption and they seek clarification on this matter. As this point is of the most interest for our journal, substantial support to this conclusion is required. Furthermore, they note that there are missing chemical formulas regarding the explanation of the carbon looping process, and they feel figures 3-6 could benefit from improved explanations.

Author's response: Thanks for the comments. We would like to emphasise that the CaL-LP scheme proposed for decarbonisation of the iron and steel industry in this study is a cyclic process, where a proportion of lime/limestone solids is continuously circulated between the two kilns to separate CO₂ from flue gas into a high-purity stream. Namely, the lime generated in the lime kiln is reused in numerous cycles to capture CO₂ in the add-on kiln. In fact, repeated lime carbonation-limestone calcination reactions (i.e., calcium looping) is such an important point for the proposed CaL-LP scheme for CO₂ capture that we use the whole sub-section entitled “Concept process description” to introduce this process integration scheme in detail, followed by the sub-section entitled “Techno-economic feasibility” to further clarify this matter. Nonetheless, in order to improve clarity, we have added the required chemical equation (Line 128, Page 6) and explanations (Lines 109–116, Page 5) in the revised manuscript. Now we believe that it is clear enough to spell out the CaL-LP scheme and avoid misunderstandings.

With regard to the requirement to improve explanations for Figures 3-6, we have improved the explanation in the context related to Figures 3-6 (Lines 201–202 and 222–223 in Page 10; Lines 227–242 in Page 11; Lines 323–329 in Page 15) of the revised manuscript. We believe it is much better now, but are still willing to improve explanations if the reviewer could provide some detailed advice/comments on the revision.

REVIEWERS' COMMENTS:

Reviewer #1 (Remarks to the Author):

The following are the revision comments on the revised manuscript:

1. The authors have added Table 1 in the Methods section, which defines the thermodynamic and techno-economic assumptions used in the study. The assumptions for techno-economic analysis should be given with a confidence interval rather than a fixed value. For example, air separation unit costs are presented as 184.8 kWh/tO₂. It is recommended to provide '±values' associated with this assumption.
2. Similar to point 1, Figure 4 and figure 6 should have a ±deviation associated with the values. The conclusion provided should be revised in accordance these values.

Reviewer #2 (Remarks to the Author):

Authors have modified the Manuscript, substantially in agreement with the comments made by this Reviewer.

Reviewer #3 (Remarks to the Author):

My contribution to the review of that manuscript focusses on the CO₂ emission reduction potential for the iron and steel industry. In that sense, the title and the abstract evoke the perception that the carbon looping process can reduce CO₂ emissions by 68% thereby exceeding set CO₂ reduction targets. Neither the title nor the abstract mention that the carbon looping process is grounded on carbon capture and storage (CCS) to achieve the CO₂ reductions. CCS has been discussed as a CO₂ reduction option especially for energy intensive industries, but progress especially in the iron and steel industry has been very little and its further development is uncertain. Considering this, the paper presents a technological option to concentrate CO₂ emissions of iron and steelmaking processes to that extent that CCS could be applied. If this is true, I am not sure if the manuscript meets the requirements for being published in Nature Communications.

Response to Referees

Reviewer #1 (Remarks to the Author):

The following are the revision comments on the revised manuscript:

1. The authors have added Table 1 in the Methods section, which defines the thermodynamic and techno-economic assumptions used in the study. The assumptions for techno-economic analysis should be given with a confidence interval rather than a fixed value. For example, air separation unit costs are presented as 184.8 kWh/t_{O₂}. It is recommended to provide '±values' associated with this assumption.

Author's response: Thank you for your advice. There are 13 parameters listed in Table 1, where a fixed value is associated with each parameter. Actually, each listed value is the most reliable and reasonable one that we have identified from various literatures currently available for this study, with the data source cited and indicated along with the value. The reason why we give a fixed value rather than a confidence interval for each parameter in Table 1 is because the values of some parameters, including the carbonation temperature, calcination temperature, fuel heat value, carbon-to-calcium ratio, economic lifetime, lime carbonation enthalpy, and limestone calcination enthalpy, have already been established in engineering practice or determined by reaction thermodynamics. However, the value of the remaining parameters in Table 1 is indeed within a confidence interval, therefore, we have performed sensitivity analysis as shown in Fig. 7, where the influence of these variable parameters on the steelmaking economics during operation of the proposed CaL-LP scheme is discussed in detail. Here in Table 1, we prefer to keep a consistent format of data, which is given in the form of a fixed value.

2. Similar to point 1, Figure 4 and figure 6 should have a ±deviation associated with the values. The conclusion provided should be revised in accordance these values.

Author's response: Thank you for your advice. Also similar to our response to Point 1, the potential deviation of the values in Fig. 4 and Fig. 6 arisen from the assumptions for techno-economic analysis in Table 1 is discussed in detail in the context related to Fig. 7.

Reviewer #2 (Remarks to the Author):

Authors have modified the Manuscript, substantially in agreement with the comments made by this Reviewer.

Author's response: Thank you very much for your time and efforts in reviewing our manuscript.

Reviewer #3 (Remarks to the Author):

My contribution to the review of that manuscript focusses on the CO₂ emission reduction potential for the iron and steel industry. In that sense, the title and the abstract evoke the perception that the carbon looping process can reduce CO₂ emissions by 68% thereby exceeding set CO₂ reduction targets. Neither the title nor the abstract mention that the carbon looping process is grounded on carbon capture and storage (CCS) to achieve the CO₂ reductions. CCS has been discussed as a CO₂ reduction option especially for energy intensive industries, but progress especially in the iron and steel industry has been very little and its further development is uncertain. Considering this, the paper presents a technological option to concentrate CO₂ emissions of iron and steelmaking processes to that extent that CCS could be applied. If this is true, I am not sure if the manuscript meets the requirements for being published in Nature Communications.

Author's response: Thank you for the comments. We fully agree with you that we should convey the information in the title and abstract that, the proposed CaL-LP scheme for decarbonisation of the iron and steel industry in this study is grounded on the wide efforts in industrial CCS. Therefore, we have modified the title and abstract accordingly in the revised manuscript.